# A Self-Configurable BUS Network Topology Based on LoRa Nodes for the Transmission of Data and Alarm Messages in Power Line-Monitoring Systems

**DOI:** 10.3390/s25051484

**Published:** 2025-02-28

**Authors:** Bartomeu Alorda-Ladaria, Marta Pons, Eugeni Isern

**Affiliations:** 1Department of Industrial Engineering and Construction, University of Balearic Islands, Ctra. Valldemossa, km 7.5, Ed. Mateu Orfila, Illes Balears, 07122 Palma, Spain; martaponsnieto@gmail.com (M.P.); eugeni.isern@uib.es (E.I.); 2Health Science and Technology Cross-Cutting Department, Balearic Islands Health Research Institute (IdISBa), 07120 Palma, Spain; 3Institute for Environmental Agro-Environmental Research and Water Economics (INAGEA), 07120 Palma, Spain

**Keywords:** wireless sensor network, power line monitoring, LoRaWAN BUS topology, Internet of Things multiprotocol

## Abstract

Power transmission lines transfer energy between power plants and substations by means of a linear chain of towers. These towers are often situated over extensive distances, sometimes in regions that are difficult to access. Wireless sensor networks present a viable solution for monitoring these long chains of towers due to their wide coverage, ease of installation and cost-effectiveness. The proposed LoRaBUS approach implements and analyses the benefits of a linear topology using a mixture of LoRa and LoRaWAN protocols. This approach is designed to enable automatic detection of nearby nodes, optimise energy consumption and provide a prioritised transmission mode in emergency situations. On remote, hard-to-reach towers, a prototype fire protection system was implemented and tested. The results demonstrate that LoRaBUS creates a self-configurable linear topology which proves advantageous for installation processes, node maintenance and troubleshooting node failures. The discovery process collects data from a neighbourhood to construct the network and to save energy. The network’s autonomous configuration can be completed within approximately 2 min. In addition, energy consumption is effectively reduced 25% by dynamically adjusting the transmission power based on the detected channel quality and the distance to the nearest neighbour nodes.

## 1. Introduction

The power grid system is one of the most critical energy infrastructure items formed by power plants, transmissions lines, sub-stations with transformers and consumers. Infrastructure ageing, electricity demand and climate change impact are addressed to ensure smart grid management, high quality service and increasing reliability requirements. In this sense, efficient monitoring and controlling systems based on information and communication technologies are introduced in power line monitoring to enhance normal operation and infrastructure damages. The use of information and communication technologies ensures robust, cost-efficient and proactive power grid operation [1,2,3]. Sensor and camera-based remote monitoring have recently gained great attention. The advantages of wireless sensor networks (WSNs) such as adaptability and scalability lead to the ability to find solutions in a variety of engineering environments, including military and defence applications [4], environmental disaster-control applications [5], monitoring of biomedical parameters affecting health [6], household applications [7] and also in what is called “precision agriculture” [8].

The use of WSNs in power grid systems offers real-time information to improve high-voltage power network monitoring and controlling to face climate change scenarios and electricity sector demands in a collaborative, cost-effective and energy-constrained way [9]. Especially for a long chain of towers covering mountain areas, wireless sensor networks (WSNs) are gaining attention due to their easy installation and high adaptability with respect to connecting different types of sensors to work collaboratively, as insect nets, monitoring a great variety of physical variables [10].

Protocols like 3G and WiFi have achieved a wide coverage area around the world, although connectivity cannot be guaranteed in some remote or non-urban areas. The lack of coverage limits the effective smart grid deployment centred on an Internet of Things (IoT) solution. To overcome this coverage constraint and expand the monitoring and controlling capabilities in remote areas, raw-LoRa and LoRaWAN protocols are gaining attention by offering a wide range of coverage with efficient energy consumption and low-cost infrastructure [9,10]. In contrast to other IoT protocols, raw-LoRa defines only the physical layer which operates in unauthorised frequency bands, and allows self-organising of the network topology like WSNs [11,12]. The raw-LoRa protocol can be configured to deploy communication solutions with one of the three most common network topologies: star, mesh or cluster tree. As shown in Figure 1, the differences between these network topologies are based on the type of nodes they include and the connection between them. In short, the nodes can be classified as “*end nodes*” which are able to measure and communicate some variable of interest, the “*router nodes*” that are used to connect end nodes located outside the coverage area with the coordinator node and the “*coordinator or gateway node*” which is the link between an end-user and the sensor network. In addition, the LoRaWAN protocol introduces the definition of the link and network layers. This establishes different roles for nodes, the ability to address messages to specific nodes, the format of messages, the use of encryption to protect the content of messages and the definition of a typical topology based on the star topology, see Figure 1.

In the case of large power tower chains across remote or mountainous areas, raw-LoRa has been identified as the most suitable protocol compared to alternatives like Narrowband Internet of Things (NB-IoT), which is more dependent on cellular mobile coverage [10]. However, the low communication speed of raw-LoRa is the main constraint in meeting the real-time requirements of power line-monitoring systems. Therefore, the combination of network technologies is a promising alternative to achieve an optimal balance between network performance of enough quality and the increase in cost that they may entail.

Compared with ordinary infrastructure monitoring located near urban areas with easy access, the monitoring challenges in mountainous and remote areas presents differences in terms of maintenance, implementation and monitoring operation [10]. In this work, a linear topology-based WSN combining raw-LoRa and LoRaWAN nodes for non-urban power transmission line monitoring is proposed. LoRaBUS is able to transmit operational monitoring data in a regular mode or generate warning messages in emergency scenarios. The complete system-monitoring scheme in mountainous areas is based on the three-layer network structure presented in [10], where raw-LoRa is selected as the collecting protocol for sensor data. This protocol offers a balanced delay time and costs results when it is combined with a cellular mobile network at the second layer to transmit sensor data to the monitoring centre.

In summary, while existing literature discusses various WSN implementations, LoRaBUS introduces several novel aspects related to topology optimisation, hybrid raw-LoRa and LoRaWAN protocol design, Emergency Transmission Priority Mode and a specific prototype for Fire Protection applications. Most prior studies focus on general WSN deployment, but LoRaBUS specifically designs a linear topology tailored for power-transmission lines. This structure improves network reliability and data-collection efficiency along a chain of towers. The combination of LoRa and LoRaWAN in LoRaBUS enhances range and network coverage while maintaining low power consumption. LoRaBUS introduces a prioritised mode for emergencies, improving the responsiveness of power line-monitoring systems. LoRaBUS goes a step further by implementing and testing a fire-protection system for remote towers, providing a tangible application of its network capabilities.

The proposed LoRaBUS topology is shown in Figure 2, formed by three types of nodes: the coordinator, the end and the router. The coordinator node is responsible for connecting the nodes of the BUS topology with a wider pre-existing network based on WAN protocols, maintaining communication between the entire network of nodes and the data-processing centre. The router node is located as the first and last node of the LoRaBUS topology as shown in Figure 2, and is responsible for translating the raw-LoRa messages from end nodes forming the BUS to the coordinator node. Therefore, the LoRaBUS topology can be connected to a wider network using one of the router nodes. If one router node is configured to connect with the coordinator node, the other router node will act as a last BUS node. And finally, the end nodes are in charge of creating the communication BUS, sending the data from the sensors and maintaining the links between them.

The main contribution of LoRaBUS is the concept and practical implementation of a robust and self-configurable tower chain communication system, which transmits sensor values during normal operation or warning messages in emergency mode. LoRaBUS uses long-range wireless communication, creating a self-configuration protocol to recognise closed nodes for easy management and fail-node recovery.

The main contributions of this paper are summarised as follows:The concept of a BUS topology in WSN with two final nodes acting as routers to improve connectivity with upper system layers.The self-configuring algorithm which detects nearby nodes and adjusts power consumption during transmission to avoid channel interference between adjacent nodes.The algorithm to transmit data between nodes avoiding data duplication and reducing power consumption.The communication architecture based on a combination of LoRa and LoraWAN for worldwide connection with controlling centre.The proposed network topology is implemented in a real prototype to test power consumption and quality coverage in a campus and non-urban scenarios.

The remaining sections of this paper are presented as follows. Section 2 discusses the network architecture by defining the different elements and connections between them. Section 3 defines the designed communication protocol and the self-configuration algorithm. Section 4 presents the experimental results obtained with the prototype in a campus and non-urban scenarios. Section 5 concludes the paper and outlines directions for future work.

## 2. System Architecture

The designed system architecture consists of three main parts: the nodes forming the BUS topology, the LoRa Network Server and the display platform where the data are represented. In this work, the LoRaBUS topology requires four different nodes: the main node (this connects the LoRaBUS topology to a coordinator node using the LoRaWAN network as shown Figure 2), the sensor nodes (represented as “end nodes” in Figure 2), the final node (special “router node” in Figure 2) and the gateway (represented as “coordinator node” in Figure 2).

The ***gateway*****node** maintains the connection between the LoRaWAN network with the network server. The data received from the BUS nodes are re-transmitted to the TTN (the global LoRaWAN community named The Thinks Network, [14]) via TCP/IP connections, such as WiFi or 3G.The ***main node*** is represented as the router node in Figure 2 and is the node placed nearest to the *gateway*. Its main function is to convert the messages from the sensor nodes (via raw-LoRa) to a LoRaWAN message that could be sent to the gateway. Furthermore, this node decides which sensor node in the BUS topology will take control of the BUS to transmit the information at each transmission cycle.The ***sensor nodes*** are able to measure the environmental conditions near their position and then send this information through the LoRaBUS implemented by themselves. Each of these nodes is therefore capable of measuring sensor data and processing an alarm message, as well as maintaining the connection between nodes to create the BUS network topology. Therefore, it is not only a node with *end node* capabilities, but is also a typical *router node* using the usual nomenclature in a WSN network. The designed LoRaBUS protocol has no restriction on the maximum number of nodes, so the total distance that could be covered by the proposed LoRaBUS topology has no theoretical limitations.The ***final sensor*** is a particular node similar to the sensor nodes which is located at the last position of the BUS; i.e., at this node the sequence of nodes in the BUS topology ends. Because the BUS communication finishes on this node, it has a different firmware although the hardware is the same as the sensors nodes. The complete functionality of this special node and other advantages are explained in Section 3.

The proposed overall system architecture described as LoRaBUS is illustrated in Figure 3, which mainly includes the different elements and the proposed connections between them.

The LoRaBUS system proposal makes use of the LoRaWAN and raw-LoRa protocols to implement the different links of the proposed topology. The LoRaWAN protocol is used to communicate between the main node and the gateway. Consequently, using the LoraWAN protocol to communicate the main node with the gateway node allows the integration of several LoRaBUS topologies into a larger LoRaWAN network. That is, the gateway node could communicate with different LoRaBUS sections through the link to the main node of each section. By contrast, the raw-LoRa protocol is used to create the BUS topology between the main node, the sensor nodes and the final node. The raw-LoRa protocol is used to build the LoRaBUS protocol described in this work, benefiting from the long-range features of the LoRa modulation. Therefore, the node accesses the radio hardware directly and messages are transmitted using LoRa modulation on the selected frequency, without message format or encryption. The next section will describe the benefits of both protocols on the LoRaBUS topology approach.

The rest of the parts of the system architecture were selected using freeware options. In this way, an open-source platform called *The Things Network* (TTN) was selected as the LoRaWAN Network Server. This platform provides easy-to-use functions to manage messages received using LoRaWAN protocols. Thus, the information from the sensors is transmitted to the gateway, which uploads it to a user-friendly display platform with a web-based GUI called Node-Red. These software solutions are used to validate the LoRaBUS topology configuration in real experiments summarised in Section 4.

### Raw-LoRa and LoRaWAN

LoRa is a wireless communication technology that was patented by *Smetech* [15] and uses a radio frequency modulation to transmit data through the ISM bands (in Europe it represents frequencies between 867 and 869 MHz). LoRa is part of the *Low-Power Wide-Area Networks* (LPWAN) and represents one of the most popular technologies in IoT projects together with Bluetooth, Zigbee or SigFox. LoRa specifications state that its coverage is up to 15 km in rural areas, and uses very low transmission rates, from 250 bps to a maximum of 5.5 kbps. In addition, this technology aims to ensure high power efficiency, which is an important feature when remote deployments are considered.

Communication through LoRa is defined in two different ways: raw-LoRa and LoRaWAN. The difference between them is related to the OSI layers that they use. For example, in a raw-LoRa mode, the communication is set point-to-point only by using the physical layer. Instead, the LoRaWAN connection defines both MAC and NET layers. Another difference is that in a LoRaWAN connection, any node needs to be in the coverage area of the coordinator or router nodes, which is unusual in remote scenarios such as forests. For that reason, we designed a prototype of communication protocols, and different types of nodes. Figure 4 represents the different types of nodes, showing the OSI layers stack used to implement each of them in the LoRaBUS.

The proposed LoRaBUS is implemented in the nodes forming the BUS topology and it uses the raw-LoRa as a physical layer as Figure 4 shows. The main node is the first node of the BUS topology and it is responsible for making the translation between data to and from the BUS nodes, from and to the LoRaWAN Gateway. While the LoRaWAN Gateway is capable of sending and receiving data from the TTN platform using Wifi connection to the Internet. The data received from the TTN platform is visualised using the Node-Red web-based GUI.

## 3. LoRaBUS Communication Protocol

In this section, the LoRaBUS communication protocol is described and explained in detail. The proposed approach tries to implement a BUS topology using LoRa-based nodes, improving the autonomy of the network. The LoRaBUS protocol is based on the implementation of a precise management of the transmission power used for sending messages between neighbouring nodes. Thus, two neighbouring nodes may be at a distance that will require a certain level of transmission power. This power must be discovered because it will depend on the distance, but also on link quality factors. Although the LoRa protocol allows management of the spreading factor which adjusts the transmission rate, receiver sensitivity and chirp rate to improve the range of the nodes, this option was not used in the proposed system because of its negative effects: increasing the time of flight of the messages and increasing battery consumption. It has been considered that a network intended for emergency alerting should not use spreading factors that reduce the data rate, increase the message flight time or reduce the battery life. Therefore, the nodes will have a spreading factor value that will not be modified by the protocol defined in this work.

The self-configuration protocol is important for the deployment of nodes in remote environments, where it is necessary to resolve situations such as the inclusion of nodes for maintenance, the addition of new nodes to the BUS or the tolerance to node failure. In cases of one node failure, the *Discovering Neighbours* process provides enough information to recover the LoRaBUS connection. The next sections will describe the protocol implemented by the LoRaBUS to create a robust and efficient approach.

### 3.1. Discovering Neighbours

When the nodes are initialised after powering on, or due to a reconfiguration process of the topology, the first stage involves self-definition and self-configuration of the linear communication BUS. This process, referred to as “Discovering Neighbours”, is based on a methodology in which each node identifies its closest neighbouring nodes on the BUS, and also the correct direction for information flow to reach the *main node* and, consequently, the user’s application.

The purpose of the “Discovering Neighbours” methodology is to construct the *nodelist* and the neighbours’ table, as depicted in Figure 5. The *nodelist* is an array containing the identifiers of all *sensor nodes* comprising the BUS, sorted sequentially from the main node to the final node. The information stored in the *neighbours’ table* of each node will be used both to establish communications between them and to prevent network cuts due to failures in nearby nodes. As can be seen in Figure 5, the table stores not only the nearest neighbour nodes, but also the following ones, so that each node has up to two nodes to maintain communication in the appropriate direction: towards the *main node* or towards the *end node*. This duplicity of neighbours introduces a certain degree of tolerance to node failures, which in outdoor environments can be due to a large number of situations. Figure 5 illustrates an example of a BUS with six nodes, where the *main node* is identified as “ID1” and the *final node* is labeled “ID6”. In this network, if node “ID2” fails then node “ID3” could attempt communication with node “ID1” based on its neighbour table. If the node that fails is ID5 then node ID3 could communicate with node ID4 and it would be “ID4” that would be in charge of reaching node “ID6” as the *final node* of the BUS topology. Thus, in order to cut off the communication, two consecutive nodes would have to fail to function correctly and then the network would be cut off at that point without the capacity to continue sending messages. The detailed procedure for obtaining the *nodelist* is described in Section 3.1.1.

On the other hand, each node in the BUS must discover and construct its neighbour’s table. This table contains the identifiers of the four nearest nodes. For example, in Figure 5, the sensor node “ID3” will identify that its closest nodes are “ID2” and “ID4”, while the next closest nodes are “ID1” and “ID5”. In addition, the Discovering Neighbours methodology is designed to define the minimum power required to transmit data between each node, i.e., to determine the lowest possible energy needed for communication. Initially, the required transmission power will be set to the minimum capable by the transmitter for the closest nodes, and to the maximum power available for the next closest nodes. This initial configuration, as shown in the table in Figure 5, will be evaluated and adjusted during the normal operation of the BUS network to determine the optimal minimal transmission power requirements for each neighbouring node. The procedure designed to obtain the *final power requirements*, the *neighbour’s table* and the *nodelist* will be explained in subsequent sections.

#### 3.1.1. Obtaining the “Nodelist”

The process to obtain the *nodelist* in a certain LoRaBUS topology can be initiated every time the end user decides, for example, in cases in which new *sensor nodes* have been introduced, or due to a network-maintenance event (battery replacement, sensor repair or other actions). The methodology to obtain a new *nodelist* is initiated by the *main node* transmitting a CONFIG message. The payload of the CONFIG message is detailed in Table 1. It contains the level of the power transmission used to send this message (*tx_power*, the list of node identifiers of the current *nodelist* that the message has passed through and a checksum based on CRC16.

The first node that sends the CONFIG message is the *main node*. The message is transmitted using just the transmission power required to reach to the next node in the topology. After that, whenever the next node receives a CONFIG message, it will check if its identifier (idx) is already included in the payload of the message. If not, it will add its identifier *idx* at the end of the list, recalculate the checksum and resend the message using the lowest transmitting power available at the transmitter. On the other hand, if a node receives a CONFIG message with its *idx* included, the node should consider the message as a confirmation that a previous sent message has been received correctly by the next node so, when it happens, the node stops retransmitting the message again. With this procedure, we reduce the duplicity of the messages in both directions of the BUS because we suppose that there will be a minimum level of power transmission which allows a node to communicate with its nearest neighbour without reaching the next closest neighbour in the BUS due to the increment in the distance between consecutive nodes. When the CONFIG message is received by the *final node*, this node will determine the complete *nodelist* by selecting the message that contains the largest number of identifiers in the *nodelist*.

Finally, with the final *nodelist* at the *final node*, a STOP message (see Table 1) is sent from the *final node* to the *main node*. In this case, the information about who is sending this message is appended at the end of the message (*idSend* in Table 1). When a *sensor node* receives an STOP message, firstly, it stores the final *nodelist*. Next, it checks who is the sender to determine if the message originated from the previous node in the *nodelist*. If so, the message must be forwarded to the next nearest *sensor node* in the direction of the *main node*. Otherwise, the message retransmission is not necessary, and it should be considered as the confirmation reception of a previous STOP message which comes from a *sensor node* nearest to the *main node*. In this case, the *sensor node* completes its configuration process and remains listening to the BUS.

Finally, when the STOP message arrives at the *main node*, all the *sensor nodes* in the BUS will continue listening the BUS, and the *nodelist* is completed and fully propagated to all nodes.

#### 3.1.2. Obtaining the Neighbour’s Table

Afterwards the *nodelist* is known for all nodes, the *Discovering Neighbours* process continues at each *sensor node* with the extraction of their nearest neighbours. Each *sensor node* creates the *neighbour’s table* with the closest and next closest nodes (four nodes at each *sensor node*). Up to four neighbours instead of just the two nearest neighbours were included to serve as a backup to automatically resolve communication conflicts, or BUS topology outages (e.g., in cases of failures of the nearest nodes). Therefore, each *sensor node* will be able to communicate with two neighbours in each direction of the BUS topology, see Figure 5. Only the two *sensor nodes* nearest to the BUS ends reduce the number of nodes included in the *neighbour’s table*.

The *neighbour’s table* is completed, initially, with the minimum transmission power observed during transmissions in the *Discovering Neighbours process*. This minimum transmission power is obtained from the payload of CONFIG and STOP messages, see details in Table 1. With the objective of improving the energy efficiency of the LoRaBUS network, the transmission power is optimised at each node to the minimum necessary to send messages to all nodes included in the *neighbour’s table*. This process is performed by one *sensor node* at a time while the rest of the nodes are only listening to the medium. The access to the medium to transmit is self-arbitrated by *sensor* nodes to reduce the interferences and retransmissions between nodes sharing the BUS.

The power of each node in the *neighbour’s table* is estimated using the procedure shown in Figure 6. The channel between nodes is tested using the specific *DISCOVERY* message (see Table 2). If the node receives a *HELLO* message with the correct *neighbour node* ID value then the level of transmission power used to send the *DISCOVERY* message is stored in the *neighbour’s table* confirming that messages reach the neighbour node using this minimum power level. At the end of this process, the final transmission power for each node in the table will be between the minimum transmission power used during the *"nodelist process"* and the maximum transmission power available at the *sensor node*. If after a timeout period, any correct *HELLO* message is received, then the *sensor node* will increase the transmission power level used in the previous *DISCOVERY* message and it will repeat the evaluation for all nodes in the *neighbour’s table* with still no response. The process finishes when the table is completed and the minimum transmission power is established for all nodes. The first node that will start the neighbour’s table will always be the *final node* after receiving the response message from the next node to the *STOP* message. This is because the *final node* is the first node which will send the *STOP* message to the rest of the nodes in the BUS, highlighting that *nodelist* is completed. The process (detailed in Figure 6) uses the *DISCOVER* messages to send from the *sensor node* that is discovering its table, while *HELLO* messages are the answers from its neighbour nodes. In both message types, the nodes include the power-transmission level used to send the message and, therefore, the receiver can store the level used by the other node.

When the node that is searching for neighbouring nodes and the minimum power requirements, has finished and completed the table, it must then send a final transmission using the *DISCOVER_NEXT* message. This special message defines the next node in the BUS that will start the process to complete its own *neighbour’s table*. The next node is defined following the *nodelist* from the *final node* to the *main node*. Therefore, the last node in completing the *neighbour’s table* will be the *main node*. If the next *sensor node* does not send its *DISCOVER* message after a timeout, the *DISCOVER_NEXT* message is resent to the next node. If the resend fails again, the *DISCOVER_NEXT* message is sent to the second nearest node, considering that the closest neighbour node is not working properly. This fail condition is reported to the *main node* for maintenance purposes. Finally, when all nodes have defined their own *neighbour’s table* the *main node* will send the *DISCOVER_END* message which will be re-transmitted to all nodes in the BUS, finishing the *Discovering Neighbours* process with the LoRaBUS configured for data communications between *sensor* nodes.

### 3.2. LoRaBUS Data Communication

The LoRaBUS configuration is proposed to create a hybrid communication protocol combining features from BUS and mesh topologies to be used in remote serial communication applications without the use of specific intermediate routers, coordinators or nodes along the line. As has been mentioned previously, the LoRaBUS is primarily proposed as an alternative for implementing a remote monitoring system near the high-voltage power lines along long distances without communication coverage. For this reason, the proposed communication protocol includes different transmission modes (stable or alarm) depending on the data value and the situation context. Therefore, the LoRaBUS approach can communicate using either of the two possible modes. In fact, LoRaBUS could be considered as a mesh network during alarm mode, and a BUS topology with arbitration during stable mode.

In stable mode, the information will be sent by each node in the network like a remote monitoring system. The stable mode is a proactive mode in which messages are sent by nodes at regular intervals. BUS access and management, in this case, is performed by the *main node* through individual requests for BUS access. The arbitration methodology used to access to the BUS is based on a combination of the well-known *master–slave* organisation, and the *token–ring* arbitration. LoRaBUS defines the *main node* as the *master* of the BUS. This means that the main node has to decide which node is going to send its information during the stable period at regular intervals. This decision is managed using a *TOKEN* message described in Table 3 where *id_token* refers to the identifier of the node that has to receive the *token*, *id_from* is the node identifier of the sender of this message and *id_to* is the identifier of the addressed node. Each node receives the *TOKEN* message at regular internals depending on the availability of the network. The use of token arbitration reduces the number of messages to be retransmitted and reduces the energy consumption due to collisions between retransmitted messages.

On the other hand, in alarm mode, the *sensor node* will start sending *ALARM* messages which includes sensor-measurement data and node identifiers (see Table 3). This mode is a reactive mode in which the nodes analyse the sensor values and decide to create and send the message in reaction to an event in process. In this case, any alarm message received by a node will be retransmitted automatically stopping any individual request for BUS usage. When this kind of message is received by any other node in the network, it will put the receiver node into alarm mode, and the message will be retransmitted to ensure it reaches the *main node*. When the *ALARM* message arrives at the end user, it must then evaluate the situation and decide whether to ignore the alert or notify emergency services. This person can also deactivate the alarm mode, allowing all nodes to return to stable operation. In *ALARM* mode, the LoRaBUS approach could be considered a mesh topology with reduced number of operations in the nodes. The message is not transmitted to all nodes in the line and is only transmitted to the nodes that allow one to reach the *main node*.

In order to achieve maximum power efficiency, the first node that will send its monitored information will be the furthest one from the gateway. This means that the final node will be the first one in sending its data to the server so it will be the first one that will receive the *TOKEN*. Following this strategy, each node in the BUS can be in a low power-consumption mode, with the function called *Deep Sleep*, a time proportional to the number of nodes which are in front of it. This provides a reliable guarantee for the long-term operation of the system.

After the reception of the *TOKEN*, the monitored data will be transmitted by the sensor node with a message of type *INFO* which includes the identifiers of the owner of those data, and the addressed node obtained from the *nodelist*. Usually the destination node will be the *main node*, but the proposal is able to allow transmissions between nodes too. The total length of the data field is considered in this work to be equal to 20 bytes, enough to include the data collected from the sensors connected to the node. The power-transmission level that will use any node would be, initially, the value saved in the neighbour’s table. Even so, if the transmitter does not detect the *INFO* message from the next node, the node will try a new transmission, increasing the transmission power level. After three failed transmissions, the node which has the token will send its information to the next neighbour available in the *neighbour’s table*, considering that the previous node was out of service.

## 4. System Test and Analysis

A proof of concept implementation was developed using one *gateway*, one *main node* and three *sensor* nodes. The last node in the BUS will work as the *final node*. All nodes and elements of the LoRaBUS network were implemented with the same transmitter equipment consisting of a Pycom LoPy 4 [16] (Pycom Ltd., Eindhoven, The Netherlands). This device is equipped with the ESP32 dual-core processor, and incorporates connectivity circuitry for WiFi, BLE and LoRa/Sigfox technology. The LoRa transmitter is built on Semtech’s SX1276 circuitry incorporating the full LoRaWAN protocol stack and capabilities to create both Class A and Class C devices. It has a 4 MB RAM capacity and is programmed using the MicroPython language.

The sensor nodes are the most critical part of the system in terms of power requirements because they are supposed to be distributed in a remote and non-controlled area so that they would have to be powered with batteries and/or solar panels. This implies assuring an efficient management of the power consumption. In addition, the *Sensor* nodes were designed to calculate some fire indexes such as the Fire Weather Index (FWI) [17]. Therefore, the sensors considered include the SEN0114 (soil moisture) (LONG WHALE FASHION UK Ltd., Kington, UK), the HPMA115S0 (particle sensor) (Honeywell International Inc., Charlotte, NC, USA), the AMG8833 (tiny thermal camera) (Adafruit Industries, LLC, New York, NY, USA), the HTU21D (temperature and humidity) (TE Connectivity Corporation, Carlsbad, CA, USA) the and BMP085 (pressure and temperature) (Adafruit Industries, LLC, New York, NY, USA). Of course, other sensors could be considered depending on the final application.

The rest of the system approach was implemented to validate not only the BUS topology, but also the connection with the end user is a realistic context. Therefore, the *main node* connects to the *gateway* using the LoRaWAN protocol. Finally, the *gateway* is registered in the Network Server TTN platform and all messages sent and received from the BUS nodes are transmitted to the Node_Red App as the final application, where the sensor data can be displayed in the user-friendly platform. The measurements can be represented in graphs, levels and warning pop-up indicators so an end user can easily interpret the data efficiently. The sensor data are visualised using the app implemented using TTN and Node_Red tools and presented in Figure 7.

This complete approach was used to analyse and evaluate the LoRaBUS approach considering its power requirements, the link quality and its performance in different scenarios.

### 4.1. Sensor Node Power-Consumption Requirements

The power consumption of a *sensor* node was measured during different stages of its workflow. The experimental results show that the highest current consumption corresponds to the LoRa transmission with a value of 160 mA. The transmission was performed with frequency = 868 MHz, tx-power = 2 dBm, bandwidth = 125 KHz, spread factor (SF) = 7 and coding rate (CR) = 4/5. In contrast, when the LoRa transceiver was turned off and CPU was in Deep Sleep mode, the consumption dropped to 7 mA. Due to a couple of issues with the LoPy design, reported by the manufacturer in [16], the CPU module draws more current than it should while in Deep Sleep. In any case, as was expected, consumption results presented in Table 4 demonstrate that the transmission task is one of the most energy-wasting processes in the entire workflow of a *sensor* node.

In contrast, when analysing the total time allocated to each task, it becomes evident that the sensor node spends most of its time idle in Deep Sleep mode (refer to the *Time* column in Table 4). Consequently, the energy consumption over a complete node cycle must consider the time dedicated to each task. Although the LoRa transmission task exhibits the highest energy consumption per unit time, the duration of this task is very short. As a result, the *Consumption* column in Table 4 shows that the energy consumption attributed to transmission is negligible compared to the energy consumed during the node’s standby periods.

These results highlight the importance of optimising both the choice of sensor devices and the CPU, particularly concerning their idle (particularly, the Deep Sleep mode) power consumption. Reducing idle power consumption is critical to significantly lowering the overall energy requirements, especially given the operational cycle anticipated for these remote monitoring systems.

An important parameter to adjust when sending a message is the transmission power level. In order to evaluate its impact on power consumption, two sensor nodes were placed 5 m apart, and the same message was transmitted while gradually increasing the transmission power level from 2 dBm (minimum) to 14 dBm (maximum). The results, shown in Figure 8, indicate that reducing the transmission power level to the minimum can decrease power consumption by 50 mA (a reduction of 25%). This behaviour is aligned with the reported work in [18], where similar IoT transmitters were evaluated reporting exponential dependence between the power consumption and the transmission power level used to send messages.

Deep Sleep mode was evaluated and tested in the proposed implementation, although the hardware used is not suitable for long periods of sleep. As for the synchronisation process between nodes, the implementation described is designed to establish long periods of communication inactivity. Therefore, it is expected that the Deep Sleep mode will be useful to maintain battery life as long as possible. The nodes in Deep Sleep mode will be synchronised by defining a similar Deep Sleep period for all nodes, and when they regain activity, a full cycle of *stable mode* will be required for success; thus, the nodes will be online until the *end node* sends its *INFO* message. After the last *INFO* message and if no *ALARM* message is received, all sensor nodes will return to Deep Sleep mode until the next period of activity. This methodology synchronises all nodes with the transmission of the last *INFO* message from *final node*.

Taking into account that the hardware used was not specifically selected for low power consumption, as the data show in Table 4, the length of time that the node will be operational will depend on the time it takes for the battery to be consumed based on its nominal level of available charge. Therefore, the node lifetime is estimated considering a battery with a capacity of 2000 mAh. The lifetime would be 2022 cycles = 7.27 days, assuming one transmission with x_power of 2 dBm every 5 min. It must be noted that the hardware used to prototype the nodes was not specifically designed for long periods of Deep Sleep function. In the current implementation, the use of renewable power sources such as solar panels is mandatory.

### 4.2. Node Communication Quality

The quality and reliability of the LoRa-based communications were also evaluated using the implemented prototype. Because the communication links use two types of LoRa-based protocols, firstly, the LoRaWAN communication between the *main node* and the *gateway* is tested and afterwards the coverage between *sensor nodes* using LoRaBUS, followed by a performance test of the proposed protocol.

The main goal of the LoRaWAN test was to evaluate three characteristic parameters of the communications, observing the influence of physical and electromagnetic interference on the communication link between the *main node* and the *gateway*. The metrics used in this test are as follows: Packet Delivery Ratio (PDR), Signal-to-Noise Ratio (SNR) and Received Signal Strength Indication (RSSI). The obstacles of this test were implemented by increasing the distance and building elements between the nodes that were continuously trying to communicate up to 300 messages every 5 min with a *payload* of 22 bytes (frequency = 868 MHZ, bandwidth = 125 kHz, SF = 7, CR = 4/5). With this configuration, the receiver has a typical sensitivity of −123 dBm. Overall, three different scenarios were evaluated:Scenario 5 m: This represents direct communication over a 5 m distance with no obstacles. The test was carried out in a laboratory on the first floor of the University of the Balearic Islands.Scenario 7 m: In this scenario, the two nodes were placed in separate rooms with a single wall between them.Scenario 20 m: This was the most complex scenario, where one node was placed 20 m away, separated by three walls and an aisle.

The RSSI results are presented in Figure 9 considering all scenarios. The x-axis is the number of messages sent from the transmitter, while the y-axis is the RSSI value expressed in dBm.

These results are coherent because when the signal propagation and received signal parameters in scenario 5 m are stronger, the RSSI is greatest (around −37 dBm). In contrast, when some obstacles are added between the nodes, the RSSI goes down to lower values below −75 dBm. Additionally to the three scenarios considered, for the scenario 20 m the time between message transmission was increased, from 5 min to 10 min. This change had as its aim exploring the impact of TTN servers on the message-processing cue. In Figure 9, the increase in time between messages during the test is observed in the RSSI values as a light reduction, explainable by a reduction in the transmission power level used by the transmitter node due to the presence of less noise in the wireless channel.

The LoRaWAN protocol automatically adjusts the power transmission needed to ensure a stable value for the SNR parameter. Again, this behaviour is the main reason explaining the low differences between scenarios reported in Figure 10 in line with the RSSI results. Most messages were received with an SNR of 6 dB, which is considered acceptable for LoRa-based communications [19].

The methodology then analyses the PDR values using the gateway to send all the messages received from the *main node* to the network server (based on the TTN platform), and then the network server sends them to the NodeRed environment, which stores all the received data. As can be seen in Table 5, the PDR value computed by sending messages every 5 min was not as high as expected, but this values cannot be explained only due to the poor quality of the communication channel because the other parameters reported satisfactory behaviour. In order to understand these results, a review of the temporary uses of the communications space was carried out to verify compliance with the recommendations for the use of the LoRa channel. In this sense, the experiment was designed to achieve a Time on Air (ToA) of 0.0566 s per message, and the transmissions were carried out at a frequency of 868 MHz every 5 min so the duty cycle was 0.02%. This means that the experiment complies with the ETSI duty cycle (<1%) [20], and the TTN Fair Use Policy (maximum uplink airtime of 30 s per day) [14]. Even so, the scenario 20 m was repeated with an interval time between messages of 10 min; as a result, the PDR obtained was 97%. Clearly, the total message received is directly related to the TTN workload.

The obtained results can be compared with the study in [21], which involved indoor experiments conducted in a nine-story building at the National Research University. According to this study, for a one-floor distance (@868 MHz, SF = 7), the average RSSI was −77.83 dBm, the SNR was 9.55 dB and the PDR was 99.8%.

### 4.3. Point-to-Point Communication

In order to evaluate the feasibility of the raw-LoRa connection in a non-controlled scenario, a *sensor node* (station) was placed in an urban area near Binissalem, a village of Mallorca (Spain), while another *sensor node* (rover) was located at different places and distances. The station node was activated to constantly transmit messages every 5 min at 14 dBm using raw-LoRa. The payload of the message was formed by an increasing counter to measure how many consecutive messages could be received at the rover node location. Figure 11 shows the position of the station node (blue point) and the different places for the rover node. The initial rover position was in another city (Inca), 7 km away from the station node.

The rover node moved from the initial point to places closer to the station node. In each location, the rover node was left 30 min to receive messages from the station node. Considering the number of consecutive messages received, the different positions were classified into three types: good quality (green), limited coverage (yellow) and out of coverage (red). From results reported in Figure 11, the distance raw_LoRa communication between nodes can be considered acceptable in the range of 1 and 2.5 km, considering urban environments and the specific hardware selected. This result aligns closely with the analysis presented in [12] by X. Zhang et al., where the authors concluded that the point-to-point communication range of LoRa is around 1km under complex environmental conditions.

### 4.4. LoRaBUS Performance Analysis

Finally, the LoRaBUS proposal was tested by means of a performance analysis that included the evaluation of the duration of node-management periods, the auto-configuration of node discovery and transmission power management, as well as the sending of messages from one end to the other of the created BUS topology network. In this performance test, three nodes were deployed on the rooftops of various buildings at the University of the Balearic Islands. The spatial distribution of these nodes is shown in Figure 12.

Once all nodes were installed in the designed location, the self-configuration stage was initialised assuming stable (no alarm) state. This process involves the Neighbour’s Discovering stage described in Section 3, after which the main node queried the monitored data from the other nodes. Table 6 summarises the time increments, following the HH:MM:SS format, required by all nodes considering the initial time as the instant of time where the *main node* sends the first *CONFIG* message. These temporal references were extracted from all transmitted and received messages, stored on SD cards at each node.

The main findings of the performance test are as follows:The *nodelist* was correctly generated in various configurations and locations. All BUS nodes successfully recognised which node identifier corresponds to the main, and final node, and also which nodes are its neighbours. Furthermore, in some locations, the *final node* directly received the *CONFIG* message from the *main node*. While this message could have been interpreted as the *nodelist*, the *final node* correctly waited for an alternative version and ultimately determined that the version forwarded by the *sensor node* was the correct one.The total time that elapsed from the moment the *main node* sent the initial *CONFIG* message to the point where all three nodes had established the *nodelist* was approximately 1 min and 10 s (from 00:02:01 to 00:03:12).The *neighbours’ table* of the *main node* and the *sensor node* included the identifiers of the other two nodes. Additionally, the *final node* correctly registered the identifier of the *sensor node*.The time required to construct the *neighbours’ table* was approximately 8 s per node, resulting in a total process duration of approximately 14 s.Following the configuration phase, the *main node* initiated a request for monitored data from the *final node* by sending a *TOKEN* message. The *sensor node* received this message and forwarded it to the *final node*. In response, the *final node* transmitted the monitored sensor data back to the *main node*, completing the process in 46 s. All communications during this phase were conducted using the minimum transmission power level.Finally, the *main node* sent a *TOKEN* to the *sensor node*, and the response was received in 17 s. The total time required to collect data from both nodes was approximately 64 s.

## 5. Conclusions

In this work, we presented a linear, LoRa-based Wireless Sensor Network approach (LoRaBUS) designed for monitoring transmission power lines. This network aims to provide a reliable communication system for electrical companies to monitor variables of interest, such as weather conditions and early forest fire detection.

Related works in the literature include the study cited in [9], where the authors propose a sensor network for monitoring overhead transmission line sag and temperature. However, their approach utilises a LoRa-Mesh topology, requiring a gateway with GSM/LTE connectivity every 4 km. The proposed LoRaBUS approach requires only one gateway on one side of the BUS, while the last node acts as a BUS termination. In the same sense, the proposed LoRaBUS system was focused on overhead transmission lines which often follow linear paths in remote areas where conventional communication technologies may not be reliable. Additionally, these infrastructures are often difficult to access, necessitating a system that is autonomous in terms of operation, maintenance and power supply. Experimental evaluations of the LoRa communication demonstrated that, with the selected hardware, sensor nodes can be spaced up to 2.5 km apart in suburban areas. However, this distance is highly dependent on the presence of obstacles between nodes. Furthermore, the network prototype was used to assess the performance of the communication protocol.

The results show that LoRaBUS creates a self-configurable linear topology that is advantageous for node-installation processes, node maintenance and node fault location. The discovery process collects data from the neighbourhood to build the network and save energy. Each node has enough information to be fault tolerant to nearby nodes and can reconfigure the BUS topology by introducing changes in transmission parameters (increasing transmission powers). The autonomous network configuration can be completed in approximately 2 min. In addition, energy consumption is effectively reduced by 25% by dynamically adjusting the transmission power based on the detected channel quality and the distance to the nearest neighbour nodes.

Future work will involve studying the impact of distribution power lines on LoRa communication, exploring other relevant low-power hardware prototypes to enhance LoRaBUS autonomy, and evaluating the benefits of small solar panels with respect to sustaining the system’s energy requirements.

## Figures and Tables

**Figure 1 sensors-25-01484-f001:**
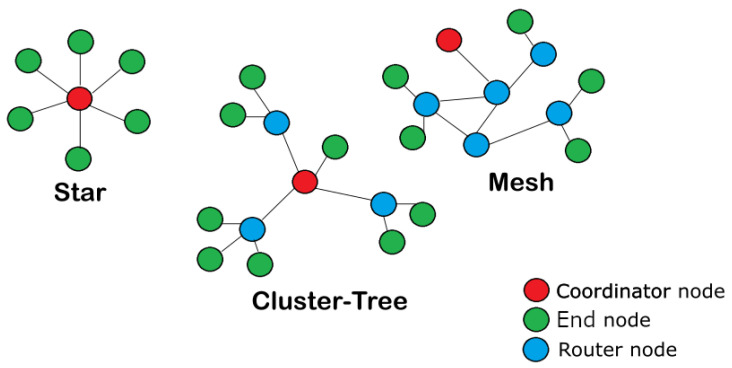
Typical topologies of WSN. Adapted from [13].

**Figure 2 sensors-25-01484-f002:**
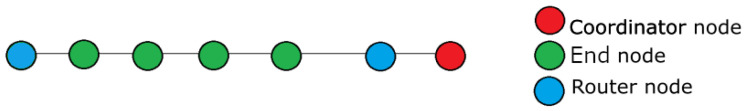
Linear topology of LoRaBUS.

**Figure 3 sensors-25-01484-f003:**
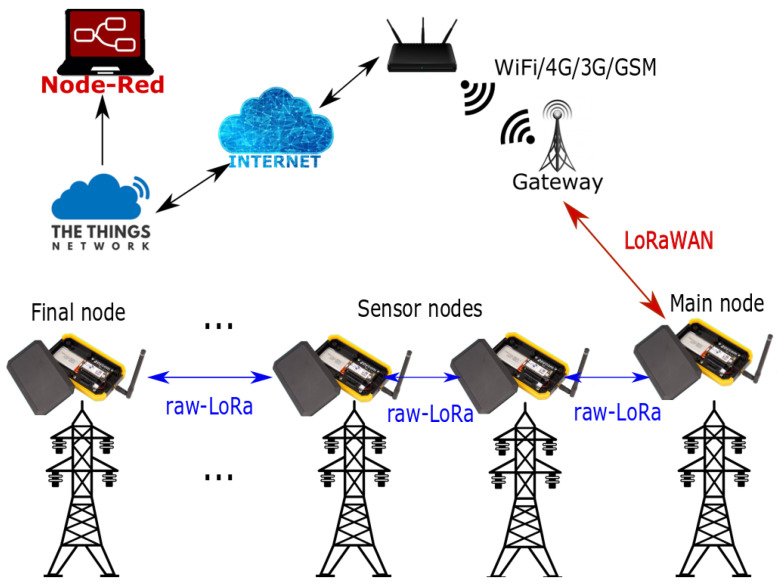
Elements of the proposed LoRaBUS system and their connections.

**Figure 4 sensors-25-01484-f004:**
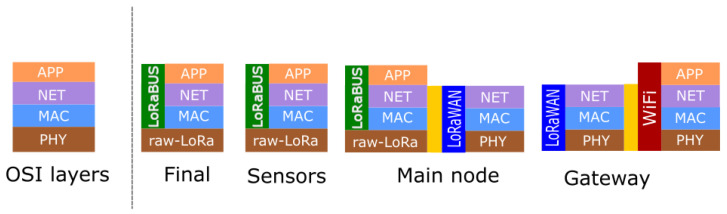
OSI layers defined in each node type.

**Figure 5 sensors-25-01484-f005:**
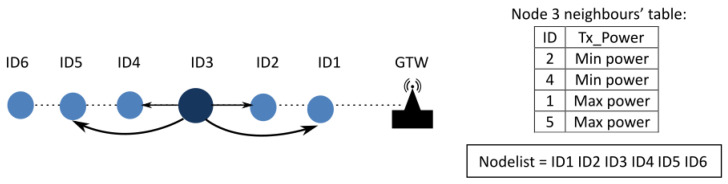
Schematic example of the results obtained with the *Discovering Neighbours* process on a 6-node BUS network: *nodelist*, *neighbour’s table* of node 3 and initial transmission power level.

**Figure 6 sensors-25-01484-f006:**
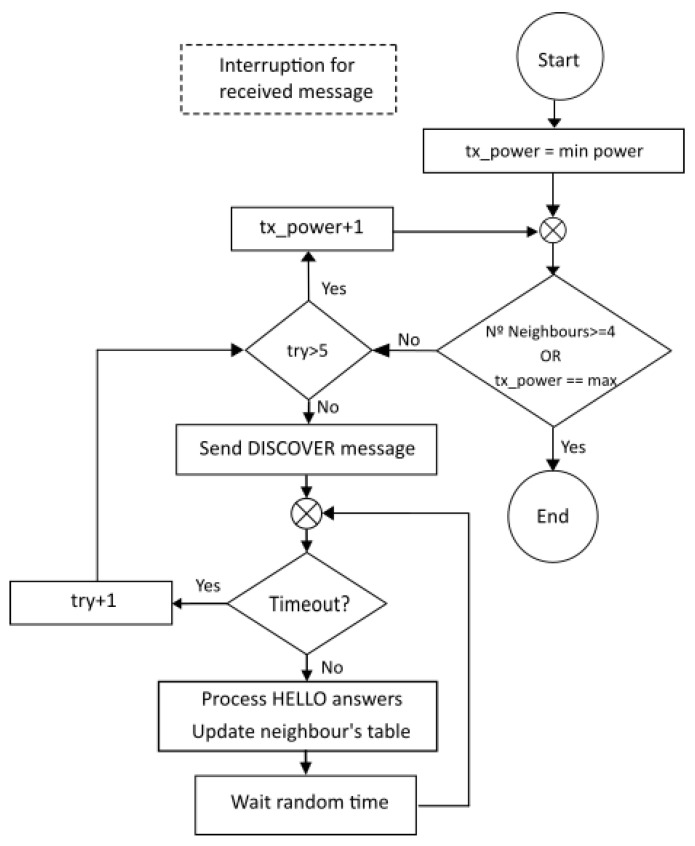
Procedure followed to complete the neighbour’s table with minimum transmission power for each node.

**Figure 7 sensors-25-01484-f007:**
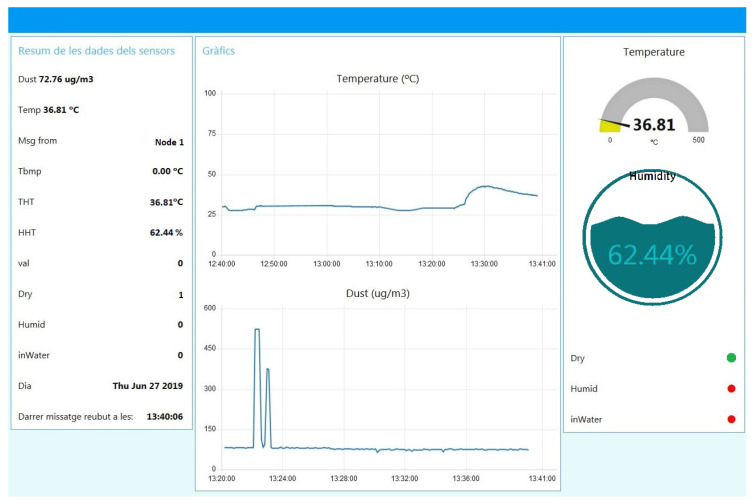
Data screen implemented on Node_Red where *sensor* node information can be reviewed.

**Figure 8 sensors-25-01484-f008:**
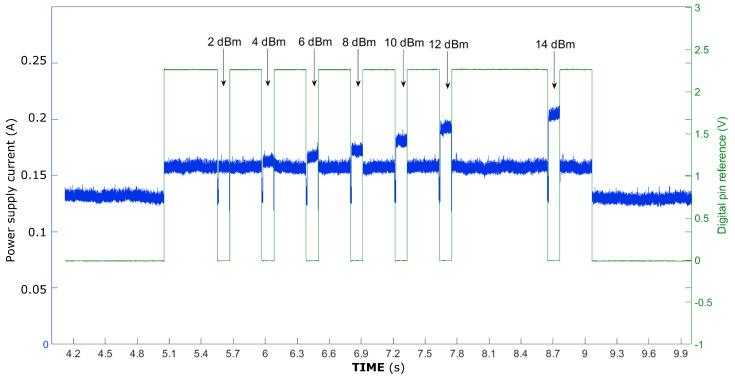
Consumption of a node with different LoRa power transmission levels.

**Figure 9 sensors-25-01484-f009:**
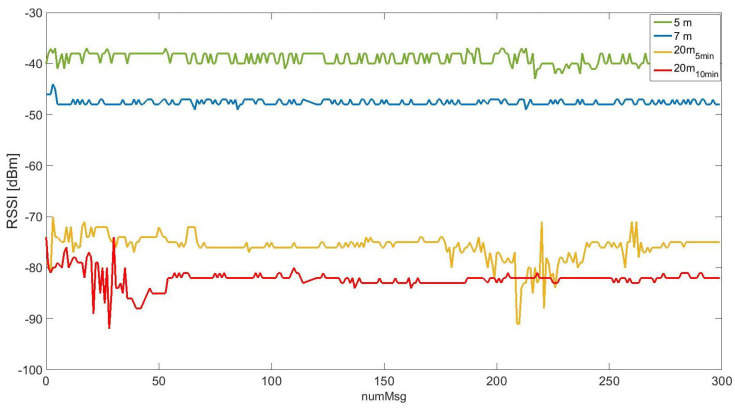
Received Signal Strength Indication (RSSI) with different distances between nodes. RSSI axis range is [−30, −100] dBm.

**Figure 10 sensors-25-01484-f010:**
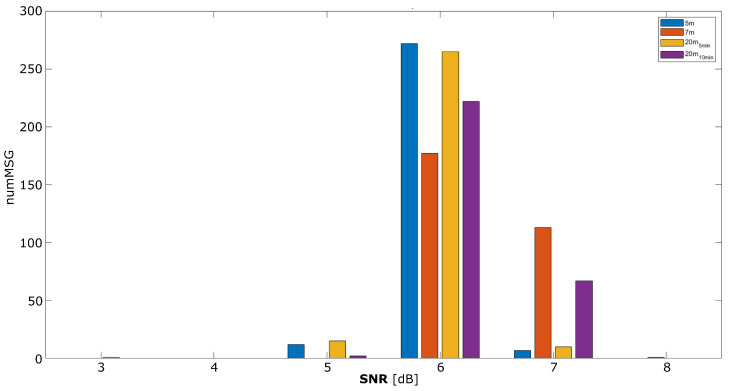
SNR for the different test scenarios. On the x-axis, we find the SNR values within the range [+3, +8] dB. On the y-axis, we find the number of messages received with each corresponding SNR value.

**Figure 11 sensors-25-01484-f011:**
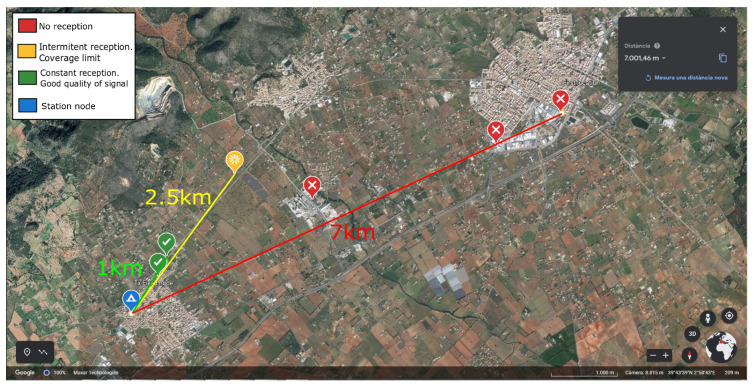
Experimental coverage test in an urban area, performed between Inca and Binissalem (Mallorca, Spain).

**Figure 12 sensors-25-01484-f012:**
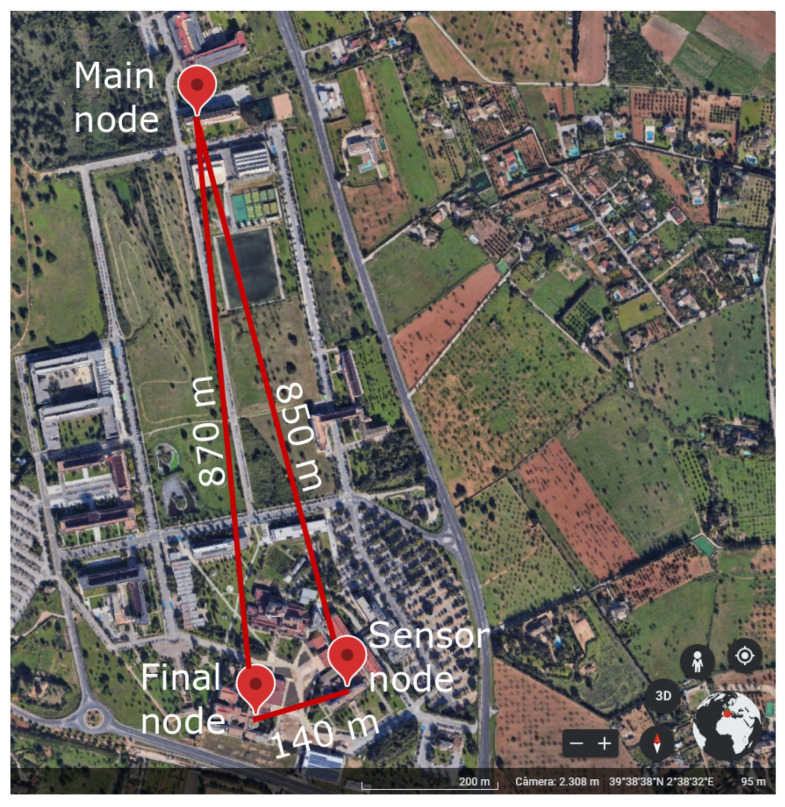
Map example of different locations used for the network operation test in the campus.

**Table 1 sensors-25-01484-t001:** Payloads of messages used in the *Discovering Neighbours* process.

Message Type	Payload
CONFIG	CONFIG + power_tx + id1 + id2 +...+ idFinal + checksum
STOP	STOP + power_tx + *nodelist* + idSend + checksum

**Table 2 sensors-25-01484-t002:** Payloads of messages used in the *neighbour’s table* process.

Message Type	Payload
DISCOVER	DISCOVER + tx_power + *id* + checksum
DISCOVER_NEXT	DISCOVER_NEXT + tx_power + *id_next* + checksum
DISCOVER_END	DISCOVER_END + tx_power + *id* + checksum
HELLO	HELLO + tx_power + *id* + checksum

**Table 3 sensors-25-01484-t003:** Payloads of messages used in the *LoRaBUS* for data transmission.

Message Type	Payload
TOKEN	TOKEN + *id_token* + *id_from* + *id_to* + checksum
ALARM	ALARM + *id_from* + *sensor_data* + checksum
INFO	INFO + *id_from* + *id_to* + *sensor_data* + checksum

**Table 4 sensors-25-01484-t004:** Summary of energy consumption of a node during transmission operation.

Mode	Current (mA)	Time (s)	Consumption (mAh)
raw-LoRa Transmitt	160	0.1150	0.0051
*Idle*	140	5	0.194
To read sensors	140	5.3	0.21
Deep Sleep	7	300	0.58
Cycle	-	310.415	0.9891

**Table 5 sensors-25-01484-t005:** Packet delivery ratio in indoor environment.

Distance (m)	Messages Sent	Messages Received	PDR (%)
5	300	231	77
7	300	230	76.67
20	300	239	79.6
20 (10 min)	300	291	97

**Table 6 sensors-25-01484-t006:** Most relevant timestamps of the workflow (timestamp format HH:MM:SS).

	Main Node	Sensor	Final
**Instant Description**	**Node**	**Node**	**Node**
Installing and placing the node	00:00:00	00:00:00	00:00:00
Sending the first CONFIG msg	00:02:01	00:02:13	00:02:15
Receiving the first CONFIG msg	00:02:14	00:02:07	00:02:13
Obtaining the *nodelist*	00:03:12	00:03:11	00:03:10
Sending DISCOVER for the 1st time	00:05:56	00:05:31	00:05:17
Obtaining its neighbour’s table	00:06:09	00:05:39	00:05:36

## Data Availability

The original contributions presented in this study are included in the article. Further inquiries can be directed to the corresponding author(s).

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
