# Peer review of "A Self-Configurable BUS Network Topology Based on LoRa Nodes for the Transmission of Data and Alarm Messages in Power Line-Monitoring Systems"

_sensors, 2025, doi:10.3390/s25051484_

Round 1

Reviewer 1 Report

Comments and Suggestions for Authors

The paper proposes LoRaBUS, a hybrid LoRa-LoRaWAN approach for node detection, energy optimization, and emergency transmission prioritization. The paper is well organized and written and the topic is very interesting. However, there some points that need further improvements. Bleow are my commenst

1.     Reflect some key numerical results in your abstract and conclusion that highlight your research findings to emphasize the impact and significance of your work.

2.     The paper lacks a dedicated Related Work section. Apart from the background provided in the introduction, there is no discussion of existing literature relevant to power line monitoring. It is essential to present the literature you reviewed before proposing this work, highlighting their strengths, weaknesses, and the gap addressed in your paper.

3.     In Section 2, System Architecture, you mention that “LoRaBUS topology requires four different nodes,” but the text only discusses three nodes, which differ from the list. Please correct it.

4.     Fault Tolerance: Have you considered node failure in your design? In other words, is your system fault-tolerant? If not, the design may lack efficiency compared to alternatives that include redundancy, especially since sensor nodes are susceptible to failure due to various factors. Elaborate on this point in the paper and discuss potential improvements.

5.     Provide more details in your paper about the DeepSleep mode and how the nodes implement it. This is important for understanding the energy optimization in your design.

6.     Clarify whether data communication occurs proactively (at regular time intervals) or reactively based on events. Elaborate on this in your paper and justify your design choice.

7.     While the work discusses recent technology, most references cited are outdated. Enrich your reference list with relevant and recent works from the last five years to demonstrate alignment with current advancements.

Author Response

The paper proposes LoRaBUS, a hybrid LoRa-LoRaWAN approach for node detection, energy optimization, and emergency transmission prioritization. The paper is well organized and written and the topic is very interesting. However, there some points that need further improvements. Below are my comments

Comments:

  1. Reflect some key numerical results in your abstract and conclusion that highlight your research findings to emphasize the impact and significance of your work.

Response: Thank you very much for the comment. Some research findings have been added in the abstract (lines 11-13) and conclusion (lines 625-633) sections.

  1. The paper lacks a dedicated Related Work section. Apart from the background provided in the introduction, there is no discussion of existing literature relevant to power line monitoring. It is essential to present the literature you reviewed before proposing this work, highlighting their strengths, weaknesses, and the gap addressed in your paper.

Response: Thank you for this comment. We have revised manuscript at lines 78-88 highlighting the existing literature relevant for communication systems on a power line monitoring system.

  1. In Section 2, System Architecture, you mention that “LoRaBUS topology requires four different nodes,” but the text only discusses three nodes, which differ from the list. Please correct it.

Response: Thank you for pointing this out. We have introduced some changes in the revised manuscript at lines 125-127 correcting the difference.

  1. Fault Tolerance: Have you considered node failure in your design? In other words, is your system fault-tolerant? If not, the design may lack efficiency compared to alternatives that include redundancy, especially since sensor nodes are susceptible to failure due to various factors. Elaborate on this point in the paper and discuss potential improvements.

Response: Yes, the design considers node failure, and a fault tolerance mechanism is already included in the section 3. During the “Discovering neighbours” methodology, each node constructs its own “neighbours’ table”. That table collect the information to send messages to the nearest nodes and the next closest nodes. Therefore, in case a sensor node fails, the next closest node is selected to overpass the failure condition and recover the LoRaBUS topology without any additional step.

The description of this methodology has been reviewed in the manuscript at lines 203-215 and 218-220 to improve the drafting of this methodology for adapting to failures

  1. Provide more details in your paper about the DeepSleep mode and how the nodes implement it. This is important for understanding the energy optimization in your design.

Response: Yes, we agree with the review that the DeepSleep mode is important for understanding the energy results. In fact, the energy consumption values obtained due to DeepSleep mode are relevant for battery life estimation. In this sense, due to the use of LoPy microcontroller, to implement the prototype, those values are higher than it is expected compared with other products. In fact, during the development of the project, we discover a note of the provider, where they explain that due to a couple of issues with the LoPy design, the module draws more current than it should while in Deep Sleep. The DC-DC switching regulator always stays in high performance mode, which is used to provide the lowest possible output ripple when the module is in use. In this mode, it draws a quiescent current of 10mA. When the regulator is put into ECO mode the quiescent current drops to 10uA. Unfortunately, the pin used to control this mode is not usable during Deep Sleep. This results in the regulator remaining in PWM mode, keeping its quiescent current at 10mA. The flash chip also doesn’t enter power down mode as the CS pin floats during Deep Sleep. This causes the flash chip to consume around 2mA of current. To solve this situation, the provider proposes to use the “Deep Sleep Shield” where the current consumption during deep sleep is between 7uA and 10uA depending on the wake sources configured. But this shield was not available for the project.

We have reported the results highlighting the quiescent current obtained in laboratory which was 7mA (line 442). Of course, lower values will provide better lifetime of battery, but also, other hardware could provide better results. In all cases, the proposed LoRaBUS topology could be applied and used independently of the final hardware used.

Some changes have been added in the manuscript explaining this issue at lines 442-445.

  1. Clarify whether data communication occurs proactively (at regular time intervals) or reactively based on events. Elaborate on this in your paper and justify your design choice.

Response: Thank you very much for pointing this out. In fact, the communication data occurs proactively during normal operation and reactively during alarm events. To Highlight this duality operation, some changes have been added in the manuscript at lines 361-372.

  1. While the work discusses recent technology, most references cited are outdated. Enrich your reference list with relevant and recent works from the last five years to demonstrate alignment with current advancements.

Response: Thank you for this comment. We have updated the references with recent works in the revised manuscript.

Reviewer 2 Report

Comments and Suggestions for Authors

The paper describes the development of a system based on LoRa radios for monitoring a linear chain of power line towers. However, some of the terminology used is questionable, and the proposed solution requires further justification.

Comments about the work:

1) The sentence in line 45 should be clarified. The LoRa protocol (LoRa PHY layer) does not inherently support any specific network topology. For networking purposes, LoRaWAN is typically used, which operates in a star topology.

2) Figure 1 is unnecessary as it illustrates the well-known topologies commonly used in Wireless Sensor Networks. Additionally, "cluster-tree" can simply be referred to as "tree."

3) In this figure and throughout the document, the term "sensor node" is used for terminal nodes. However, "sensor node" typically refers to any node in a WSN. For end devices, the term "end node" is more appropriate.

4) On page 3, the "LoRaBus topology" is introduced as being based on a Bus topology. However, as far as I understand, a Bus topology is a network configuration where all nodes are connected to a single central communication line. In wireless systems, this means that all nodes share the same communication channel, being within the same communication range. This does not align with the presented work, as the nodes are not within the same range, and no justification is provided for defining it as a Bus topology.

I understand the use of a linear topology, as it aligns with the configuration of transmission power line towers. However, the way the nodes are connected in Figure 2 does not reflect what is described by the communication protocol outlined in Section 3. In this figure, there is a gateway, a router node located near the gateway, and several sensor nodes. In line 293 the authors state: “The LoRaBUS configuration is proposed to create a BUS topology to be used in remote serial communication applications without routers ...”. This statement is not correct. In reality, the presented system includes a gateway, an end node (the terminal node), and all the other nodes function as routers.

At first glance, the network appears to follow a tree topology, with each router having only one branch (linear topology). However, based on the communication protocol description, it is evident that the actual topology is mesh. The use of “node tables” with four nodes to ensure redundancy in case of node failure emphases the implementation of a mesh topology with alternative paths. This topology is more complex than the star topology used in LoRaWAN. Therefore, all this description should be clarified in the work.

5) Furthermore, given that many protocols have been developed for WSN with mesh purposes, is the presented system the most suitable for this application? Additional justification is needed to support this choice.

6) In section 3.1.1 where the “nodelists” are created, if several nodes can receive the CONFIG packet and resend the message how collisions are avoided?

7) What guarantees that the CONFIG and STOP messages will not be lost?

8) The authors refer that "If the node receives a HELLO message with the correct neighbour node id  value then the level of transmission power is stored in the neighbour’s table because the both  nodes can send messages using this minimum power level." This may not be true due to the asymmetry of the propagation channel.

9) The authors rely on transmission power to reach sensor nodes at longer distances. However, one of the key features of LoRa technology is the use of the spreading factor parameter to achieve extended range. However, the presented protocol does not take this into account, raising questions about whether the optimal choices have been made.

10) In Section 3.2, it is mentioned that the nodes can enter Deep Sleep mode. How are the sensor nodes synchronized to determine the appropriate time interval for entering sleep mode?

11) In section 4, system text and analysis, is missing the specifications of the system components (LoRa radios, antenna gain, software to program radios, etc).

12) Is missing the receiver sensitivity of the receiver in RSSI analysis.

13) The example used to test the system, shown in Figure 12, is limited, as it only involves three nodes. It is also important to test the system in a scenario that more closely resembles the real-world application of power line monitoring. If this is not feasible, the development of a test system that approximates this scenario should be considered, along with a better analysis of the system performance, mainly the proposed topology.

Author Response

The paper describes the development of a system based on LoRa radios for monitoring a linear chain of power line towers. However, some of the terminology used is questionable, and the proposed solution requires further justification.

Comments about the work:

1) The sentence in line 45 should be clarified. The LoRa protocol (LoRa PHY layer) does not inherently support any specific network topology. For networking purposes, LoRaWAN is typically used, which operates in a star topology.

Response: Yes, we agree with the reviewer. The manuscript has been updated in lines 44 – 59 adjusting the clarity of the sentence.

2) Figure 1 is unnecessary as it illustrates the well-known topologies commonly used in Wireless Sensor Networks. Additionally, "cluster-tree" can simply be referred to as "tree."

Response: We agree with the reviewer that Figure 1 is unnecessary for advanced readers, but we have introduced in order to show the different node roles in WSNs. Additionally, because the Tree and the cluster-tree topologies are similar, the main difference is to show that the cluster-Tree consists in the figure in the join of three starts topologies, where each router node collect the data from each node in the start and connect with the coordination node. This is quite similar to the topology proposed, where the LoRaBUS could be part of a Cluster-Tree, where the coordinator connect with the router nodes using LORAWAN, while each local network topology could be start, mesh, three, or LoRaBUS, combining topologies and protocols. That is the reason why we introduce the concept of Cluster-Tree, but if the editor agrees with the reviewer, this figure could be removed from the final manuscript.

3) In this figure and throughout the document, the term "sensor node" is used for terminal nodes. However, "sensor node" typically refers to any node in a WSN. For end devices, the term "end node" is more appropriate.

Response: Although the reviewer is correct is the assumption that the “sensor node” is the “end node” in WSNs, in the work, the LoRABUS has one “End node” or “Final node” which is the responsible to finish the message propagation in the BUS topology. In addition, all nodes have sensing functionality and can start messaging in alarm mode. For all these reasons, the name “End node” could be confusing in the LoRaBUS topology proposal.

In order to clarify the names used, the section 2 where the different names are introduced has been reviewed according to the reviewer comment.

4) On page 3, the "LoRaBus topology" is introduced as being based on a Bus topology. However, as far as I understand, a Bus topology is a network configuration where all nodes are connected to a single central communication line. In wireless systems, this means that all nodes share the same communication channel, being within the same communication range. This does not align with the presented work, as the nodes are not within the same range, and no justification is provided for defining it as a Bus topology.

I understand the use of a linear topology, as it aligns with the configuration of transmission power line towers. However, the way the nodes are connected in Figure 2 does not reflect what is described by the communication protocol outlined in Section 3. In this figure, there is a gateway, a router node located near the gateway, and several sensor nodes. In line 293 the authors state: “The LoRaBUS configuration is proposed to create a BUS topology to be used in remote serial communication applications without routers ...”. This statement is not correct. In reality, the presented system includes a gateway, an end node (the terminal node), and all the other nodes function as routers.

At first glance, the network appears to follow a tree topology, with each router having only one branch (linear topology). However, based on the communication protocol description, it is evident that the actual topology is mesh. The use of “node tables” with four nodes to ensure redundancy in case of node failure emphases the implementation of a mesh topology with alternative paths. This topology is more complex than the star topology used in LoRaWAN. Therefore, all this description should be clarified in the work.

Response: Yes, the reviewer has corrected assumptions. Thank you for your comments, we have updated the manuscript trying to clarify the topology description. We have used “End node” in the section where a generic topology is used, and the name “Sensor node” for the specific implementation of the LoRaBUS. The Figure 2 has been updated including the Final and the Main node as routers node using traditional WSN names. The manuscript has been updated to highlight this additional justification in lines 125 – 145

5) Furthermore, given that many protocols have been developed for WSN with mesh purposes, is the presented system the most suitable for this application? Additional justification is needed to support this choice.

Response: We agree with the reviewer that the LoRaBUS topology is a mix between some features from mesh networks and some others from BUS topology. In fact, LoRaBUS could be considered as a mesh network during Alarm mode, and a BUS topology with Token arbitration during Stable mode to reduce the number of messages to be retransmitted and to reduce the power consumption due to collisions between retransmitted messages.

The LoRaBUS approach generate a mesh topology reducing the number of operations in the nodes, because the message is not transmitted to all nodes in the line. The messages are only transmitted to the nodes that allow to reach the ‘Main Node’.

The manuscript has been updated to highlight this additional justification in lines 334 – 335.

6) In section 3.1.1 where the “nodelists” are created, if several nodes can receive the CONFIG packet and resend the message how collisions are avoided?

Response: This question is answered inside section 3.1.1. In line 268, it is explained that the process is initiated by the Main node and when a node receives a CONFIG message, it read if its identification data is included, in that case the message is not resend. The collisions are avoided because due to the methodology used, only the node which receives a CONFIG message with its identification data not included in the message, will transmit again the CONFIG message including its identifier. The reviewer has to take into account that only one node is working transmitting a message, while the rest of the nodes are listening to the channel. To clarify this methodology, the text of this section has been reviewed. From line 232 to line 246.

7) What guarantees that the CONFIG and STOP messages will not be lost?

Response: Thank you to the reviewer for this question. It is not well explained in the manuscript. We have changed the text of section 3.1.1 from line 268 to line 270 in order to explain how the system react in case CONFIG or STOP message will lost.

8) The authors refer that "If the node receives a HELLO message with the correct neighbour node id value, then the level of transmission power is stored in the neighbour’s table because both nodes can send messages using this minimum power level." This may not be true due to the asymmetry of the propagation channel.

Response: The reviewer introduces the asymmetry of the propagation channel, but in the paper each node transmits and evaluate the minimum power level required to reach to the next neighbour node. Of course, if the channel features are not symmetric, the minimum power level will be different for each node. In our experiments we have not observed differences, but of course, if it successes, the methodology described will decide different minimum transmission power for each transmission direction, because each channel between node is tested in both directions.

We have updated the manuscript from line 317 to line 324 to clarify this behaviour.

9) The authors rely on transmission power to reach sensor nodes at longer distances. However, one of the key features of LoRa technology is the use of the spreading factor parameter to achieve extended range. However, the presented protocol does not take this into account, raising questions about whether the optimal choices have been made.

Response: The reviewer is right; we have written in the paper about transmitter power because the Spread Factor has been left unchanged throughout the paper. It is true that the SF parameter is used to achieve a larger coverage range of a LoRa transmitter. But changing this parameter will both reduce the data rate and increase the time of flight of each message. Both effects are not desirable in a system where the main action is to provide Alarm information for decision making. The SF have been selected at minimum possible in order to maintain highest data rate and shortest time-on-air of messages. This explanation has been introduced in the reviewed manuscript in lines 231 – 238.

10) In Section 3.2, it is mentioned that the nodes can enter Deep Sleep mode. How are the sensor nodes synchronized to determine the appropriate time interval for entering sleep mode?

Response: The Deep Sleep mode has been evaluated and tested in the proposed implementation, although the hardware used is not suitable for long sleeping periods. Regarding to the synchronization between nodes, the application that is described in the paper will establish long periods of communication inactivity. Therefore, it is expected that Deep Sleep mode will be useful to remain battery life as longer as possible. The methodology proposed to synchronize the nodes is based on a definition of a similar Deep Sleep period for all nodes, and when they recover the activity, a complete stable mode cycle will be required to success, so, the nodes will be on-line until the Final Node send its INFO message. After the last INFO message and no ALARM messages are received, all Sensor nodes will return to the Deep Sleep mode until next activity period. This methodology synchronizes all nodes with the transmission of the last INFO message. This explanation has been introduced in the reviewed manuscript in lines 469 – 479.

11) In section 4, system text and analysis, is missing the specifications of the system components (LoRa radios, antenna gain, software to program radios, etc).

Response: We agree with the reviewer, these details have not included in the first manuscript version. In the revised manuscript, the section 4, lines 408 – 414, has been modified to include enough details of these specifications.

12) Is missing the receiver sensitivity of the receiver in RSSI analysis.

Response: Although the receiver sensitivity is missing, the Spread Factor was reported. Both parameters are related. But also, we agree with the reviewer and receiver sensitivity has been included in line 502.

13) The example used to test the system, shown in Figure 12, is limited, as it only involves three nodes. It is also important to test the system in a scenario that more closely resembles the real-world application of power line monitoring. If this is not feasible, the development of a test system that approximates this scenario should be considered, along with a better analysis of the system performance, mainly the proposed topology.

Response: We agree with the reviewer that a more realistic scenario is always desirable for the evaluation of the proposals, but in this case the large distances between nodes make realistic monitoring while maintaining adequate distances very difficult. On the other hand, the proposed system is based on three nodes: one acts as the main node, one as a sensor node and the last one as the final node. Additional sensor nodes could be included in the testing of the system, but the authors believe that no additional information would be obtained from a pilot with more sensor nodes. The complete behaviour of the proposed LoraBUS protocol has been tested and verified using the proposed configuration. In this sense, a laboratory prototype with 5 sensor nodes was used to verify a more complex scenario, but with an unrealistic distance between nodes that has not been added in the article due to its irrelevance beyond the functional validation of the proposal. The proposed protocol introduces an arbitration mechanism that allows scaling the solution with minimal impact on node performance, remembering that each node only keeps the table of the nearest neighbour nodes, therefore, the number of nodes has not been observed to be relevant in the applicability of the protocol.

Reviewer 3 Report

Comments and Suggestions for Authors

Authors attempt to monitor Power Lines with LoRa-based Wireless Sensor Networks. The LoRaBUS system, proposed in this work, consider a linear topology mixing LoRa and LoRaWAN protocols. LoRaBUS provides an automatic detection procedure of nearby nodes, an optimization of energy consumption, and a prioritized transmission mode in emergency situations. A propotype fire protection system was implemented and tested.

- Authors should include more recent papers in their related works.

- Authors should adapt the nomenclature of nodes in their LoRaBUS between Figures 2 and 3.

- Authors must define TTN in the first time it appears in the text. 

- Authors must review the text, searching for typos and language problems.

- There are incomplete phrases in the text, for example, "When the minimum power of the four nodes in the table is established."

- In Section 3.1.2, what does it means when the transmission power is incremented ("Sensor node will increase the transmission power level")?

- Why the authors did not tested some well-known ad-hoc routing algorithms, such as the AODV, instead of their proposed discovery procedure?

- In Table 4, authors must make clear that the values of Current [mA] arefor transmission.

- In the last paragraph of Section 4.1, authors must define the batterylifetime. It seems to be the time until the energy of the battery is finished.

- The main application discussed in this work if the Power Line Monitoring System. However, the authors did not perform tests in this specific scenario.In this specific scenario, there should be high interference values, because of the proximity of power lines. Authors did not consider this expected interference for the evaluation of the quality of links. Moreover, since communication is performed by LoRa, some investigation could be performed, regarding other configurations, for example, with respect to the spreading factor values.

Comments on the Quality of English Language

- Authors must review the text, searching for typos and language problems.

- There are incomplete phrases in the text, for example, "When the minimum power of the four nodes in the table is established."

Author Response

Authors attempt to monitor Power Lines with LoRa-based Wireless Sensor Networks. The LoRaBUS system, proposed in this work, consider a linear topology mixing LoRa and LoRaWAN protocols. LoRaBUS provides an automatic detection procedure of nearby nodes, an optimization of energy consumption, and a prioritized transmission mode in emergency situations. A propotype fire protection system was implemented and tested.

- Authors should include more recent papers in their related works.

Response: Thank you for this comment. We have updated the references with recent works in the revised manuscript.

- Authors should adapt the nomenclature of nodes in their LoRaBUS between Figures 2 and 3.

Response: We agree with the reviewer, the manuscript and the nomenclature has been adapted to the LoRaBUS approach.

- Authors must define TTN in the first time it appears in the text. 

Response: We agree with the reviewer, the manuscript has been updated

- Authors must review the text, searching for typos and language problems.

Response: The manuscript has been revised finding some typos and language problems that has been changed. Thank you.

- There are incomplete phrases in the text, for example, "When the minimum power of the four nodes in the table is established."

Response: The phrase has been reviewed, and the manuscript has been changed. Thank you very much.

- In Section 3.1.2, what does it means when the transmission power is incremented ("Sensor node will increase the transmission power level")?

Response: The transmission power is one of the parameters of the transceiver that can be adjusted to increase or reduce the coverage distance of the message. In our approach, this parameter is used in the raw-LoRa protocol to create the BUS topology. Managing the transmission power, each node can control which neighbour node will receive the message, reducing the interferences with nodes in other parts of the network and reducing the energy consumption during transmission.

- Why the authors did not tested some well-known ad-hoc routing algorithms, such as the AODV, instead of their proposed discovery procedure?

Response: The Ad hoc On-Demand Distance Vector routing is a well-known routing protocol for mobile ad hoc networks and other Wireless Sensor Networks, where the number of connections between the nodes could increase with the number of nodes in the network. In AODV protocol, each node maintains a routing table like in the proposed approach, but the definition of the routing table is complex. Each row of the routing table has the information about the route to achieve the destination and other parameter like active neighbour for that route. But in our implementation the BUS topology simplifies enormously the discovery process. The application of AODV is not suitable and introduce complexity for a static network with just two directions to communicate with all nodes. That is the reason why a new discovery procedure is defined using a raw-LoRa transmitter as a hardware resource.

- In Table 4, authors must make clear that the values of Current [mA] are for transmission.

Response: we agree with the reviewer and the caption of table 4 has been changed

- In the last paragraph of Section 4.1, authors must define the battery lifetime. It seems to be the time until the energy of the battery is finished.

Response: we agree with the reviewer, that the capacity of the battery has been considered equal to the nominal capacity indicated by the manufacturer. When the battery arrives to this nominal capacity usage, the nominal voltage and current values decrease rapidly and therefore the battery is considered discharged. In this sense, to calculate the lifetime expected of the battery, the nominal capacity could be used as a maximum value. Of course, if a solar recharge system is considered, the recharge cycle can extend the period of nominal operation of the node. We have reviewed the section 4.1 to clarify this.

- The main application discussed in this work if the Power Line Monitoring System. However, the authors did not perform tests in this specific scenario. In this specific scenario, there should be high interference values, because of the proximity of power lines. Authors did not consider this expected interference for the evaluation of the quality of links. Moreover, since communication is performed by LoRa, some investigation could be performed, regarding other configurations, for example, with respect to the spreading factor values.

Response: we agree with the reviewer that interferences from power lines could be expected. These interferences may reduce the maximum distance between nodes or introduce retransmissions between nodes. In this work, those effects are not considered because can be adjusted additionally to the proposed topology discovery process. In addition, the distance between nodes should be as far as possible, but this distance is not a relevant feature to create the proposed topology. Of course, it has to be considered in a final installation. The authors have planned to continue with experiments with real power lines towers which due to regulatory issues is complex to access to the locations. The results reported in this work, help to validate the feasibility to design a simple and efficient BUS topology considering raw-LoRa transceivers.

Round 2

Reviewer 1 Report

Comments and Suggestions for Authors

The authors have addressed my concerns. I accept the paper in its current form

Comments on the Quality of English Language

it is perfect 

Author Response

Thank you very much for your review report,

Reviewer 2 Report

Comments and Suggestions for Authors

In Figure 2, the terms “end nodes” and “router nodes” are interchanged. A router can receive, transmit, and retransmit messages. End nodes only transmit messages. In case of figure 2, with the exception of the coordinator and the final node, all other nodes are routers.

On pp 410 and 443, there is a question mark in the number of references.

Author Response

In Figure 2, the terms “end nodes” and “router nodes” are interchanged. A router can receive, transmit, and retransmit messages. End nodes only transmit messages. In case of figure 2, with the exception of the coordinator and the final node, all other nodes are routers.

Response: Thank you for this comment. We agree with the reviewer's definition of router nodes. In the linear topology described in Figure 2 we have included two ‘router nodes’ because these nodes have similar functions and are the two ways to connect to the ‘coordinator node’. That is, we have assumed that the LoRaBUS bus can connect to the LoRAWAN network using one of the two ‘Router Nodes’ described in Figure 2. When the ‘Router Node’ is used to connect to the Coordinator node, the other end is used as an ‘End Bus Node’. This behaviour could be interchanged between both nodes depending on which one is used to connect to the ‘Co-ordinator Node’.

Because both ‘Router Node’ are not just an ‘End Node’, considered as the node that is connected to the sensors and that connects to the neighbouring nodes, Figure 2 represents these differences by extending the meaning of ‘End Node’. We want to keep this figure as it is originally included, but it could be rearranged if considered by the editor.

On pp 410 and 443, there is a question mark in the number of references.

Response: Thank you for this comment. The question mark have been corrected in the manuscript.

Reviewer 3 Report

Comments and Suggestions for Authors

Authors provided quite a substantial improvements in this paper, in comparison with the previous version of the manuscript. There is just a minor issue to be addressed: "The first paragraph of Sections 4 and 4.1 lacks the reference [?]."

Author Response

Authors provided quite a substantial improvements in this paper, in comparison with the previous version of the manuscript. There is just a minor issue to be addressed: "The first paragraph of Sections 4 and 4.1 lacks the reference [?]."

Response: Thank you for this comment. The question mark have been corrected in the manuscript.